

# Effects of combined aerobic and resistance training on glycemic control, blood pressure, inflammation, cardiorespiratory fitness and quality of life in patients with type 2 diabetes and overweight/obesity: a systematic review and meta-analysis

Sameer Badri AL-Mhanna[1], Alexios Batrakoulis[2], Wan Syaheedah Wan Ghazali[1], Mahaneem Mohamed[1], Abdulaziz Aldayel[3], Maha H. Alhussain[4], Hafeez Abiola Afolabi[5], Yusuf Wada[6], Mehmet Gülü[7], Safaa Elkholi[8], Bishir Daku Abubakar[9] and Daniel Rojas-Valverde[10]

[1] Department of Physiology, School of Medical Sciences, Universiti Sains Malaysia, Kelantan, Malaysia
[2] Department of Physical Education and Sport Science, School of Physical Education, Sport Science and Dietetics, University of Thessaly, Trikala, Greece
[3] Exercise Physiology Department, King Saud University, Riyadh, Saudi Arabia
[4] Department of Food Science and Nutrition, College of Food and Agricultural Science, King Saud University, Riyadh, Saudi Arabia
[5] Department of General Surgery, School of Medical Sciences, Universiti Sains Malaysia, Kelantan, Malaysia
[6] Department of Zoology, Ahmadu Bello University, Zaria, Nigeria
[7] Department of Sports Management, Faculty of Sport Sciences, Kirikkale University, Kirikkale, Turkey
[8] Department of Rehabilitation Sciences, College of Health and Rehabilitation Sciences, Princess Nourah bint Abdulrahman University, Riyadh, Saudi Arabia
[9] Department of Human Physiology, Federal University Dutse, Jigawa, Nigeria
[10] Centro de Investigación y Diagnóstico en Salud y Deporte, Escuela Ciencias del Movimiento Humano y Calidad de Vida Universidad Nacional de Costa Rica, Heredia, Costa Rica

Corresponding authors
Sameer Badri AL-Mhanna, sameerbadri9@gmail.com
Wan Syaheedah Wan Ghazali, syaheeda@usm.my

## ABSTRACT

**Background:** Structured aerobic or resistance training alone seems to be a beneficial tool for improving glucose homeostasis, chronic systemic inflammation, resting cardiovascular function, and mental health in people with obesity and type 2 diabetes mellitus (T2DM). The aim of the present study was to synthesize the available data on the effectiveness of combined aerobic and resistance training (CART) on glycemic control, blood pressure, inflammation, cardiorespiratory fitness (CRF), and quality of life (QoL) in overweight and obese individuals with T2DM.
**Methods:** A database search was carried out in PubMed, Web of Science, Scopus, Science Direct, Cochrane Library, and Google Scholar from inception up to May 2023. The Cochrane risk of bias tool was used to assess eligible studies, and the GRADE method to evaluate the reliability of evidence. A random-effects model was used, and data were analyzed using standardized mean differences and 95% confidence intervals. The study protocol was registered in the International Prospective Register of Systematic Reviews (ID: CRD42022355612).

**Results:** A total of 21,612 studies were retrieved; 20 studies were included, and data were extracted from 1,192 participants (mean age: 57 ± 7 years) who met the eligibility criteria. CART demonstrated significant improvements in body mass index, glycated hemoglobin, systolic and diastolic blood pressure, C-reactive protein, tumor necrosis factor-alpha, interleukin-6, CRF, and QoL compared to ST. These findings highlight the significance of exercise interventions such as CART as essential elements within comprehensive diabetes management strategies, ultimately enhancing overall health outcomes in individuals with T2DM and overweight/ obesity.No differences were found in resting heart rate between CART and ST. An uncertain risk of bias and poor quality of evidence were found among the eligible studies.

**Conclusion:** These outcomes show clear evidence considering the positive role of CART in inducing beneficial changes in various cardiometabolic and mental health-related indicators in patients with T2DM and concurrent overweight/obesity. More studies with robust methodological design are warranted to examine the dose-response relationship, training parameters configuration, and mechanisms behind these positive adaptations.

## INTRODUCTION

Obesity and type 2 diabetes mellitus (T2DM) are the most prevalent chronic metabolic diseases worldwide and are linked to cardiometabolic complications resulting in an augmented risk of metabolic syndrome, cardiovascular disease, and stroke (*Badri Al-mhanna et al., 2024a*; *Bhupathiraju & Hu, 2016*). T2DM is influenced by several risk factors. Firstly, obesity is recognised as a major risk factor for T2DM, as excess body fat can lead to insulin resistance. Secondly, physical inactivity is considered a significant risk factor, with sedentary lifestyles contributing to insulin sensitivity issues. Furthermore, unhealthy dietary habits, characterized by high sugar and fat intake, are also implicated in the development of T2DM (*Badri Al-mhanna et al., 2024b*).

Thus, populations with unhealthy weight and T2DM are likely to demonstrate physical inactivity, raised blood pressure levels, impaired lipid profile, poor antioxidant capacity, and attenuated mental health (*de Wit et al., 2009*; *Liu et al., 2022*). Obesity is highly associated with T2DM due to the increased abdominal and intra-abdominal fat distribution and increased intrahepatic and intramuscular triglyceride content that has been reported as a critical risk factor for developing T2DM, causing insulin resistance and β-cell dysfunction (*Gilyana et al., 2024*). Interestingly, the cost of obesity-related comorbidities has been estimated at US dollars 2.0 trillion globally, indicating a major problem for healthcare systems nowadays (*Afolabi et al., 2023*; *Tremmel et al., 2017*). Thus, the documentation of effective behavioral interventions appears to be a high priority for

public health policy makers to enhance awareness of the vital role of regular exercise among the masses.

Regular exercise has been reported as a foundational piece of the preventive, management, and treatment puzzle for individuals struggling with obesity and T2DM (*Colberg et al., 2016*; *Kanaley et al., 2022*; *Kemps et al., 2019*; *Kim et al., 2019*; *Melguizo-Ibáñez et al., 2023*; *Umpierre et al., 2011*; *Zhao et al., 2021*). In addition, exercise for populations with lifestyle-related chronic diseases is considered one of the most popular trends in the global health and fitness industry (*AL-Mhanna et al., 2022a*; *Kercher et al., 2023*). According to the exercise prescription guidelines by the American College of Sports Medicine, combined aerobic and resistance training (CART) is recommended for overall health improvement, even without weight loss, in people with metabolic health impairments (*American College of Sports Medicine et al., 2021*). More specifically, CART seems to be the optimal training modality for inducing beneficial changes in several cardiometabolic health-related indicators in adults with overweight/obesity, but without comorbidities (*Batrakoulis et al., 2022a*; *Yapici et al., 2023*). Furthermore, aerobic and/or resistance training can positively affect individuals with T2DM, exhibiting positive alterations in glycemic control, cardiovascular function, chronic inflammation, and mental health (*Güngör et al., 2024*; *Kanaley et al., 2022*; *Mendes et al., 2016*; *Pesta et al., 2017*; *Qadir et al., 2021*; *Sabag et al., 2017*; *Zhao et al., 2021*). However, CART is superior to aerobic or resistance training alone in improving glycemic control in this population (*Church et al., 2010*; *Jamka et al., 2022*; *Sigal et al., 2007*).

CART has been shown to improve insulin sensitivity and glucose uptake by skeletal muscles. Resistance training increases muscle mass and glycogen storage (*Knuiman, Hopman & Mensink, 2015*), while aerobic exercise enhances mitochondrial function and glucose utilization (*Wang et al., 2020*). Together, they promote better glycemic control by reducing insulin resistance and improving glucose metabolism (*Yavari et al., 2012*).

In terms of inflammatory markers, CART can lead to decreased levels of pro-inflammatory cytokines like TNF-alpha and IL-6, while increasing anti-inflammatory markers such as IL-10 (*Annibalini et al., 2017*). Moreover, cardiovascular function benefits significantly from CART. Aerobic exercise enhances cardiac output, vascular function, and endothelial health (*Al-Mhanna et al., 2023*), while resistance training improves arterial stiffness and blood pressure regulation (*Okamoto, Masuhara & Ikuta, 2013*). CART combines these benefits, leading to improved cardiovascular fitness, and heart health enhancement. Therefore, the superiority of CART over single-mode exercise likely stems from its ability to capitalize on the unique effects of both aerobic and resistance training, resulting in broader and more profound improvements in cardiometabolic health-related indicators (*Jorge et al., 2011*).

However, the effectiveness of various CART protocols still needs to be determined (*Kadoglou et al., 2013*). Hence, further investigation is necessary, since no comprehensive scientific evidence is currently available with respect to the beneficial role of CART in several health-related parameters of T2DM patients with excessive weight.

Although it is well documented that physical exercise is vital for reducing cardiometabolic risk in obesity and T2DM (*American College of Sports Medicine et al., 2021*), the effectiveness of CART in order to achieve this particular benefit is not clear. Thus, the aim of this systematic review and meta-analysis was to evaluate the effects of CART across a broad spectrum of cardiometabolic health-related parameters in people with overweight/obesity and T2DM, such as glycemic control, blood pressure, chronic inflammation, cardiorespiratory fitness (CRF), and quality of life (QoL).

## METHODS

### Registration

This systematic review and meta-analysis were conducted in accordance with the Preferred Reporting Items for Systematic Reviews and Meta-Analyses statement guidelines (*Page et al., 2021*). The study protocol was registered in the International Prospective Register of Systematic Reviews (ID: CRD42022355612).

### Literature search strategy

We obtained articles from PubMed, Web of Science, Scopus, Science Direct, Cochrane Library, and Google Scholar through a systematic electronic search. Four authors (M.H.A., A.B.D., W.S.W.G., and H.A.) utilised a set of keywords and Boolean operators, including "OR" and "AND", to conduct this electronic literature search from the beginning of the databases up to May 1, 2023, as outlined in *Al-Mhanna et al. (2023)*. The search utilised specific keywords ("Diabetes") AND ("Exercise" OR "Training"), as detailed in Table S1, to identify relevant literature. The search strategy involved employing keywords aligned with the PICOS framework, encompassing: (P) Population: individuals with type 2 diabetes mellitus (T2DM) who are overweight or obese; (I) Intervention: cardiovascular and resistance training (CART); (C) Comparator: various exercise modalities, no exercise, or standard treatment (ST); (O) Outcomes: glycated haemoglobin (HbA1c), systolic (SBP) and diastolic blood pressure (DBP), resting heart rate (RHR), C-reactive protein (CRP), tumour necrosis factor-alpha (TNF-$\alpha$), interleukin-6 (IL-6), cardiorespiratory fitness (CRF), quality of life (QoL), and body mass index (BMI); and (S) Study type: randomised controlled trials (RCTs) and controlled clinical trials. Additionally, reference lists of included articles were scrutinised to identify studies meeting the inclusion criteria, along with reference lists of pertinent systematic reviews.

### Eligibility criteria

Studies meeting the following criteria were considered eligible for inclusion:
(i) participants were diagnosed with T2DM and were concurrently overweight (BMI 25–29.9 kg/m2) or obese (BMI ≥ 30 kg/m2); (ii) there was no specified age limit for participants; (iii) the intervention used in the studies was CART; (iv) studies investigated at least one of the following primary outcomes in humans: glycemic control (HbA1c), resting cardiovascular function (SBP, DBP, and RHR), chronic inflammation (CRP, TNF-$\alpha$, IL-6), and physical fitness (CRF). Mental health (QoL) and anthropometry (BMI) were also

considered as secondary outcomes due to their association with various cardiometabolic health-related indices; (v) articles needed to be fully accessible in text and published in a peer-reviewed journal from inception up to May 1, 2023; (vi) there were no language restrictions; and (vii) studies were either randomised controlled trials (RCTs) or controlled clinical trials. Studies were excluded if they met any of the following criteria: (i) involved a mixed sample of individuals (*e.g.*, apparently healthy individuals, non-diabetic individuals with overweight/obesity, or overweight/obese individuals without T2DM); (ii) effects of CART could not be isolated due to exercise training being part of a multifaceted intervention (*e.g.*, diet and exercise intervention); (iii) the control group also performed exercise; (iv) did not assess the specified outcome measures; (v) involved an acute exercise intervention (*e.g.*, single bout or duration ≤2 weeks); or (vi) were review articles, case reports, studies lacking a control group, or had ambiguous or unclear data as previously described in *Al-Mhanna et al. (2023)*.

## Study selection

Four authors (A.B., M.G., A.B.D., and H.A.) utilised a linear appraisal method to evaluate eligibility criteria. They reviewed titles, abstracts, and full texts (in cases of uncertainty) and rigorously assessed the remaining articles against the criteria before reaching decisions. In instances of disagreement or uncertainty, a fifth author (S.B.A.L.) provided assistance, employing the same method independently. The literature search records were managed using EndNote X9 software from Clarivate Analytics, Philadelphia, PA, USA, as detailed in *Al-Mhanna et al. (2023)*.

## Data extraction

Two authors (A.B. and A.B.D.) conducted independent data sampling and extraction from relevant studies following a thorough review of the full texts. The studies included in the analysis provided substantial data encompassing details such as the primary author, publication year, demographics (population and gender), sample size, specifics of exercise interventions (frequency, intensity, duration, type), study duration, and outcome measures. Any uncertainties or discrepancies were resolved by a third author (S.B.A.L.) to maintain consistency in the data selection and extraction process. This meticulous approach aimed to enhance the reliability and transparency of our meta-analysis by minimising errors and ensuring robust data extraction procedures.

## Risk of bias assessment

Two authors (S.B.A.L. and A.B.D.) evaluated the risk of bias using the criteria described in the Cochrane Handbook (*Higgins et al., 2019*). Each study's bias risk was assessed based on specific factors, including: (i) random sequence generation; (ii) allocation concealment; (iii) blinding of participants and personnel; (iv) blinding of outcome assessors; (v) completeness of outcome data; (vi) selectivity of outcome reporting; and (vii) other biases (Table S2). Three levels of risk bias (*e.g.*, high, concerns, and low) were used to classify eligible studies as described in *Al-Mhanna et al. (2023)*.

## Data analysis

We performed all analyses using Review Manager 5.4 software (Cochrane Collaboration, available at https://revman.cochrane.org/info). We applied a random-effects model to present the outcomes and assessed heterogeneity using Cochran's Q test and the $I^2$-test. If $I^2$ was ≥50%, we used a fixed-effects model to calculate pooled results and conducted a subgroup analysis. Effect sizes were calculated using mean differences (MD) or standardised mean differences (SMD) with 95% confidence intervals (CI). Statistical significance was defined as a two-sided $p$-value < 0.05. We used the GRADEpro methodology (available at https://www.gradepro.org) to evaluate the reliability of evidence, categorising studies as low-, moderate-, or poor-quality evidence (Table S3).

## RESULTS

### Literature search and selection

From the selected databases (PubMed, Web of Science, Scopus, Science Direct, Cochrane Library, and Google Scholar), a total of 21,612 studies were initially identified (Fig. 1). After eliminating duplicate articles, the number of unique studies eligible for further assessment was reduced to 19,965. Following a review of titles and abstracts against predetermined inclusion and exclusion criteria, 19,926 studies were excluded. Subsequently, the full text of the remaining 39 articles was meticulously examined, leading to the exclusion of 14 articles with specific reasons. Hence, 25 records were selected for this study. However, among them, five reports represented follow-up studies of trials that met the criteria and were also included in this analysis. Consequently, a total of 20 studies were ultimately incorporated into this study, with data collected from 1,192 eligible patients (Fig. 1 and Tables S4 and S5).

### Literature characteristics

The studies have been categorised based on the World Bank's classification of economies for analytical purposes into different income groups (*Bank, 2024*). Fifteen out of the 20 trials were from high-income countries (*Annibalini et al., 2017*; *Church et al., 2010*; *Cuff et al., 2003*; *Dunstan et al., 1998*; *Earnest et al., 2014*; *Ferrer-García et al., 2011*; *Gibbs et al., 2012*; *Hale et al., 2022*; *Lambers et al., 2008*; *Loimaala et al., 2003*; *Magalhães et al., 2019*; *Maiorana et al., 2001*; *Scheer et al., 2020*; *Sigal et al., 2007*; *Tessier et al., 2000*), two trials were from upper-middle-income countries (*Jorge et al., 2011*; *Tan, Li & Wang, 2012*), and three trials from lower-middle income countries (*Sabouri et al., 2021*; *Yavari et al., 2012*; *Zarei et al., 2021*). Only one (*Ferrer-García et al., 2011*) out of the 20 trials was conducted in a non-clinical setting (home-based intervention). Eight studies conducted short-term exercise interventions lasting 8–12 weeks and 12 studies conducted long-term exercise interventions lasting 16–60 weeks (Table 1).

### Risk of bias assessment results

Figure 2 illustrates the summary of the risk of bias assessment. Figure 3 shows the judgements per domain for each eligible study in detail. In particular, the large majority of
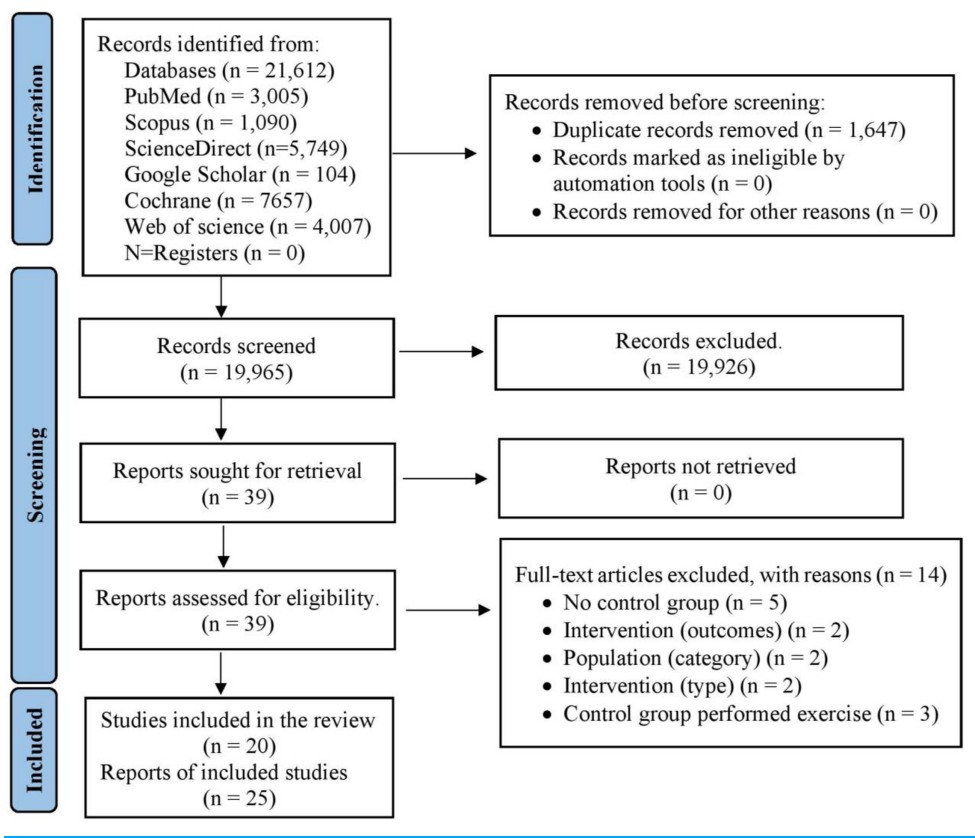

**Figure 1 PRISMA flowchart for search strategy.**

eligible studies indicated some concerns. However, a low risk of bias in missing outcome data and the selective reporting process was detected in the majority of eligible studies.

## Primary outcomes

### Glycemic control

HbA1c was reported in 16 trials involving 742 participants and showing low-quality evidence. CART demonstrated a reduction in HbA1c (SMD −0.37, 95% CI [−0.60 to −0.13]; $I^2$ = 55%; $p$ = 0.003) (Fig. 4 and Table S3) compared to ST.

### Resting cardiovascular function

RHR and SBP were included in two trials ($n$ = 65) and 10 trials ($n$ = 695), respectively, demonstrating very low-quality evidence. No differences were found in both RHR (SMD −0.65, 95% CI [−1.59 to 0.29]; $I^2$ = 59%; $p$ = 0.17) and SBP (SMD −0.04, 95% CI [−0.41 to 0.32]; $I^2$ = 80%; $p$ = 0.81) between CART and ST (Figs. 5 and 6, Table S3). Instead, CART demonstrated a significant reduction in DBP (SMD −0.33, 95% CI [−0.63 to 0.04]; $I^2$ = 60%; $p$ = 0.03) reported in 10 trials ($n$ = 560) with very low-quality evidence) (Fig. 7 and Table S3) compared to ST.

### Inflammation

CRP, TNF-α, and IL-6 were reported in three ($n$ = 167), two ($n$ = 71), and two ($n$ = 40) trials, respectively. No meaningful change was reported in CRP (SMD −0.02, 95% CI

AL-Mhanna et al. (2024), *PeerJ*, DOI 10.7717/peerj.17525

**Table 1 Characteristics of included trials.**

| Reference | Participants' age, BMI & population | Comorbidities | Study design | Recruitment & grouping | Control intervention | Test-group intervention & context | Duration | Outcome measures | Pro-instrument measure |
|---|---|---|---|---|---|---|---|---|---|
| 1. Church et al. (2010) | 55.8 ± 8.7 years, | Cancer, Neuropathy, Myocardial infarction, Heart catheterization, and coronary artery bypass surgery | RCT | Over media, mailers, and community events $N = 117$: CO = 41: Ex = 76 | Continued with their normal daily activities | 10 kcal/kg per week expending aerobic and twice a week. resistance training. Supervised | 36 weeks | 1. BM1 | 1. Maximal estimated metabolic equivalent tasks (METs) calculation |
| 2. Swift et al. (2012) | 35.8 ± 6.2, | | | $N = 96$ CO = 37 EX = 59 | | | | 2. HbA$_{1c}$ and CRP | 2. Automated glycosylated hemoglobin analyzer (DCA 2000, Bayer, Dublin, Ireland) |
| 3. Ferrer-García et al. (2011) | Inter-racial Sedentary men and women in the USA | _ | RCT | By clinical interview $N = 84$: CO = 40: Ex = 44 | Dietary and exercise counseling | 45 min of moderate physical (particularly aerobic) for 5 days + Strength training at least 2 days/week Home-based physical education program (HPEP) (verbally and in writing recommendation) | 24 weeks | 1. Quality of life 2. HbA$_{1c}$ 3. Anthropometri: BMI (kg/m$^2$), waist circumference (cm) | 1. Health-Related Quality of Life (HRQL) questionnaire with analogic scale 2. High performance liquid chromatography (HPLC) 3. – |
| 4. Maiorana et al. (2001) | | | RCT | Hospital recruitment $N = 16$: Ex = 8 CO = 8 | They were instructed not to undertake any formal exercise or change in their habitual physical activity levels during these period | 1 h circuit training (a combination of cycle ergometry, treadmill walking and resistance training) Supervised by an experienced exercise physiologist | | 1. HbA$_{1c}$ | 1. HPLC system (Hercules, CA) |
| 5. Maiorana et al. (2002) | 66.7 ± 8.0 years, | | | | | | 8 weeks | 2. Vo$_{2max}$ 2. BMI 3. Resting heart rate | 2. Vo2max, Sensormedics 3. – 4. – |
| 6. Lambers et al. (2008) | 31.3 ± 6.2 | | RCT | Hospital outpatients $N = 54$: CO = 16: Ex = 19 | Continued with their normal daily activities | 40-sessions of circuit training (walking or jogging, elbow flexion and extension, cycling, knee flexion and extension, stepping, cooling down + stack weight Strengthening Exercises Supervised | 12 weeks | 1. HbA$_{1c}$ 2. Peak oxygen consumption 3. Fitness 4. BMI, and blood pressure 5. Quality of life | 1. HPLC (Pierce Chemical Co., Rockford, IL, USA) 2. Oxycon Pro spirometer (Jaeger–Viasys Healthcare, Hoechberg, Germany 3. 6 min walk test (6MW T), sit-to-stand test 4. Digital balance, stadiometer, measuring tape and manual sphygmomanometer 5. General Health Survey Short Form (SF-36) |

| Reference | Participants' age, BMI & population | Comorbidities | Study design | Recruitment & grouping | Control intervention | Test-group intervention & context | Duration | Outcome measures | Pro-instrument measure |
|---|---|---|---|---|---|---|---|---|---|
| 7. *Hale et al. (2022)* | | | Randomized, two-arm, parallel, open-label trial | *Via* general practices (GP), Diabetes NZ, public media advertising, and health agencies that work with Maori and Pacific communities $N = 169$: CO = 84: Ex = 85 | Usual care | Diabetes Community Exercise Programme (DCEP) twice-weekly. It consists aerobic exercise warm-up (5 min), an aerobic and resistance exercise circuit with a focus on major muscle groups (30 min), and flexibility exercises (5 min). supervised | 12 weeks | 1. HbA$_{1c}$ 2. Body mass index and blood pressure 3. Fitness 4. Quality of life | 1. – 2. – 3. Incremental Shuttle Walk Test (ISWT), NZ Physical Activity Questionnaire-Short Form (NZPAQ-SF) 4. Audit of Diabetes-dependent Quality of Life (ADDQoL) Questionnaire, and the EuroQol five dimensions questionnaire (EQ-5D-5 L) |
| 8. *Loimaala et al. (2003)* | | Hypertension | RCT | Through newspaper advertisement $N = 49$: CO = 25: Ex = 24 | Conventional treatment only | Conventional treatment together + heart ratecontrolled endurance training twice a week + muscle strength training twice a week Supervised | 60 weeks | 1. VO$_{2max}$, 2. Heart rate variability (HRV) 3. HbA$_{1c}$ | 1. Respiratory gas analyzer (Sensormedics 2900Z; Sensormedics BV, Bilthoven, the Netherlands). 2. ECG with phenylephrine method for the BRS 3. Immunoturbidimetric method (Roche, Basel, Switzerland) with Cobas Mira Plus and Cobas Integra automatic analyzers |
| 9. *Tessier et al. (2000)* | Spanish patients | | RCT | Hospital outpatients $N = 39$: CO = 20: Ex = 19 | Continue with their usual activity regimen | Three times a week of rapid walk, strengthinig and strtching exercise Supervised | 16 weeks | 1. VO$_{2max}$ 2. HbA$_{1c}$ 3. Quality of life | 1. Balke–Naughton treadmill protocol 2. QOL and attitudes toward DM 3. QOL and attitudes toward DM questionnaire |
| 10. *Dunstan et al. (1998)* | 52 ± 2 years, | | RCT | Non-regular vigorous exercising NIDDM volunteers $N = 27$: CO = 12: Ex = 15 | No formal exercise | 60 min circuit weight training (CWT) 3 days a week supervised | 8 weeks | 1. Blood pressure | 1. Dinamap 1846SX automatic blood pressure measuring device (Critikon, Tampa, Florida, USA) |
| 11. *Jorge et al. (2011)* | 29.6 ± 3.4 | | RCT | From Diabetes Outpatient Clinic $N = 22$: CO = 12: Ex = 12 | No formal exercise | 3 days per week of cycling at the heart rate corresponding to the lactate threshold (30 min) plus 7-exercise circuit as follows: leg press, bench press, lat pull down, seated rowing, shoulder press, abdominal curls, and knee curls (30 min). supervised | 12 week | 1. BMI 2. IL-6 3. HbA$_{1c}$ | 1. Anthropometric scale 2. Enzymee immunoassay kits 3. Immunoassay by turbidimetry on Dimension RXL Max SIEMENS equipment and HOMA-insulin resistance index (HOMA-IR) |

(Continued)

AL-Mhanna et al. (2024), *PeerJ*, DOI 10.7717/peerj.17525

| Reference | Participants' age, BMI & population | Comorbidities | Study design | Recruitment & grouping | Control intervention | Test-group intervention & context | Duration | Outcome measures | Pro-instrument measure |
|---|---|---|---|---|---|---|---|---|---|
| 12. *Oliveira et al. (2012)* | | | RCT | Diabetic clinic recruitment N = 22: CO = 12: Ex = 10 | No formal exercise | Cycling 5 min, two sets of 15 reps strength exercise and another 5 min of cycling Supervised | 12 weeks | 4.Blood pressure 5.Vo$_{2max}$ | 4. Aneroid sphygmomanometer 5. Fit Mate Cosmed, Rome, Italy system and lactate threshold |
| 13. *Sigal et al. (2007)* | Australia | | RCT | Recruited through advertising, physicians, and word of mouth N = 126: CO = 62: Ex = 64 | Reverted to pre-study exercise levels. | Three times weekly progressive treadmills or bicycle ergometers exercised + 7 progressing weight machines resistant exercises Supervised | 22 weeks | 1. HbA$_{1c}$ 2. Plasma lipid values, and blood pressure 3. BMI | 1. Turbidimetric immunoinhibition 2. Enzymatic methods on a Beckman-Coulter LX20 analyzer 3. Bioelectrical impedance analyzer, computed tomography (CT). |
| 14. *Reid et al. (2010)* | | | RCT | Hospital recruitment N = 109: CO = 52: Ex = 57 | Revert to their pre-study activity level | Three times per week treadmills and/or bicycle ergometers + progressing weight lifting exercise Supervised | 22 weeks | 4.QoL | 4. Medical Outcomes Trust Short-Form 36-item version (SF-36) |
| 15. *Gibbs et al. (2012)* | 55.8 ± 9.7 years | Hypertension | RCT | Recruited *via* newspaper advertisements from greater Baltimo N = 140: CO = 70: Ex = 70 | Information about American Heart Association Diet with no further intervention | Three times per week of 10–15 min warm-up, 45 min of aerobic exercise at 60–90% maximum heart rate and a cool down +7 weight training exercises of latissimus dorsi pull down, leg extension, leg curl, bench press, leg press, shoulder press, and seated mid-rowing as two sets of 12–15 repetitions at 50% of 1–repetition maximum. supervised | 24 weeks | 1. VO2peak 2. BMI, blood pressure 3. Body and visceral fat | 1. Modified Balke protocol 2. Kgm$^{-1}$, automated BP device (Dinamap MPS Select; Johnson & Johnson, New Brunswick, NJ) 3. Dual X-ray absorptiometry, magnetic resonance imaging |
| 16. *Yavari et al. (2012)* | 28.9 ± 2.8 | | RCT | DM Clinic out-patients N = 40: CO = 20: Ex = 20 | Maintained their lifestyle | Three times per week of a warm-up stage, they worked for 20–30 min on a treadmill or bicycle plus 2 sets each of 8 exercises with 8–10 repetitions on weight machines. supervised | 52 weeks | 1. HbA$_{1c}$ 2. VO2max | 1. Auto-analyzer devices (Hitachi®, model 704, 902, made in Japan) 2. Rockport 1,600 m walking test |

| Reference | Participants' age, BMI & population | Comorbidities | Study design | Recruitment & grouping | Control intervention | Test-group intervention & context | Duration | Outcome measures | Pro-instrument measure |
|---|---|---|---|---|---|---|---|---|---|
| 17. *Cuff et al. (2003)* | | | RCT | Hospital recruitment N = 19: CO = 9: Ex = 10 | | Warm-up, an aerobic phase, a resistance training phase, and a cooldown to total a class time of 75 min. supervised | 16 weeks | 1. Peak VO$_{2max}$ 2. Abdominal adipose tissue & mid-thigh skeletal | 1. Progressive Naughton protocol treadmill test 2. Computed tomography scans |
| 18. *Earnest et al. (2014)* | From two general practice centers in the Netherlands and Belgium | | RCT | N = 117: CO = 41: Ex = 76 | Maintained their lifestyle | Treadmill walking 3–5 days per week at a moderate to vigorous intensity + 2 days of strength training exercises (abdominal, upper & lower body exercises) for 145 min supervised | 36 weeks | 1. Estimated METs 2. HbA$_{1c}$ 3. Waist circumference, body composition | 1. From VO$_2$Peak and time-to-exhaustion (TTE) 2. Beckman Coulter DXC600 Pro (Brea, CA) 3. – |
| 19. *Scheer et al. (2020)* | | | RCT | Recruited from the community using local media advertising. N = 35: CO = 21: Ex = 14 | Maintained usual activities | Three times a week in a heated community pool of eight aerobic stations in alternating with eight resistance stations | 8 weeks | 1. VO2peak 2. BMI | 1. Mass flow ventilometry, and simultaneous mixing chamber analysis of expired gas fractions (Vmax, Sensormedics, Yorba Linda, USA) 2. Body weight (AND HW 200KGL scales, Australia) and height |
| 20. *Tan, Li & Wang (2012)* | 62.7 ± 15.3 years | | RCT | Recruited *via* local medical practitioners. N = 30: CO = 12: Ex = 18 | Maintain their usual physical activity habits | Three sessions per week of warm-up period (30 min), moderate aerobic exercise, resistance training (10 min) with five leg muscle exercises (two sets of 10–12 repetitions) and a cool-down supervised | 24 weeks | 1. Body composition 2. BMI, 3. HbA$_{1c}$ | 1. GE Prodigy direct digital DEXA bone densitometry (GE Healthcare, USA) 2. Kgm$^{-2}$, Waist girth was measured at the level of the umbilicus horizontally without clothing, while the hip girth at the level of the greatest protrusion of the gluteal muscles with underwear 3. By enzymatic method (BioRad, Hercules, USA). |
| 21. *Zarei et al. (2021)* | | | RCT | N = 26: CO = 13: Ex = 13 | Received no intervention | Three sessions per week of aerobic exercise (walking or running) + weight training | | 1. BMI 2. HbA$_{1C}$ | 1. Body composition analyzer (InBody 270, South Korea) 2. Pars Azmoon kit (Pars Azmoon Co, Tehran, Iran) and biochemical auto-analyzer device |

*(Continued)*

AL-Mhanna et al. (2024), *PeerJ*, DOI 10.7717/peerj.17525

Peerj

| Reference | Participants' age, BMI & population | Comorbidities | Study design | Recruitment & grouping | Control intervention | Test-group intervention & context | Duration | Outcome measures | Pro-instrument measure |
|---|---|---|---|---|---|---|---|---|---|
| 22. *Magalhães et al. (2019)* | New Zealand communities | Hypertension | RCT | Using media advertisements and e-mail<br>N = 38<br>CO = 22<br>EX = 16 | No intervention | Aerobic exercise on cycling at 40% to 60% of the heart rate reserve (HRR) + RT included one set of 10–12 repetitions. | 1 year | 1. BMI<br>2. Fasting glucose<br>3. HbA$_{1c}$<br>4. VO$_{2MAX}$ | 1. BMI (kg/m$^2$)<br>2. –<br>3. –<br>4. – |
| 23. *Magalhães et al. (2020)* | 53.3 ± 5.1 years | – | | CO = 27<br>EX = 28 | | | | 5. Inflammatory Markers | 1. Commercial ELISA kits. |
| 24. *Annibalini et al. (2017)* | 29.3 ± 3.7<br>Italy | – | RCT | —<br>N = 16<br>EX = 8<br>CO = 8 | Usual care (no intervention) | Aerobic exercise performed on a treadmill with (40% to 65% of heart rate (HR) reserve) and duration (30 to 60 min) + RT gradually increased from 2 to 4 sets of 20 to 12 repetitions from 40% to 60% of 1-repetition maximum (1-RM) for 3 times per week | 16 weeks | 1. BMI<br>2. HbA$_{1c}$,<br>3. VO$_{2MAX}$ | 1. BMI (kg/m$^2$)<br>2. –<br>3. – |
| 25. *Sabouri et al. (2021)* | 69.3 ± 4.2 years<br>Iran | – | RCT | N = 28<br>EX = 15<br>CO = 13 | Subjects in the CO group were asked to continue routine activities without participating in any exercise program throughout the study. | Aerobic exercise on cycle ergometers + RT for 70 min were performed three training sessions/week | 12 weeks | 1. BMI<br>2. HbA$_{1c}$<br>3. DBP and SBP | 1. Body composition analyzer (In Body 570, Korea)<br>2. –<br>3. – |

**Note:**
RCT, Randomised control trial; EX, Exercise; CO, Control.

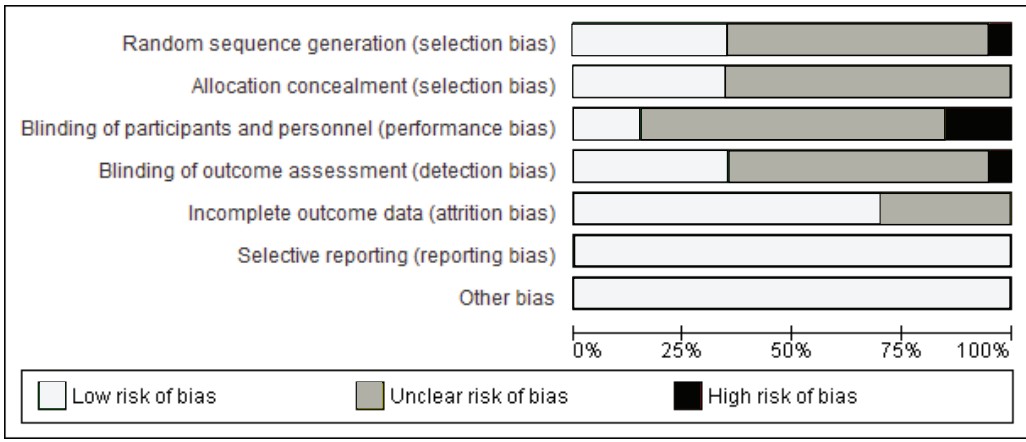

**Figure 2  Summary of the risk of bias.**

[−0.32 to 0.29]; $I^2$ = 0%; $p$ = 0.92) with moderate-quality evidence between CART and ST. CART group exhibited reductions in TNF-α (SMD −0.96, 95% CI [−1.52 to −0.39]; $I^2$ = 12%; $p$ = 0.009) and IL-6 (SMD −0.94, 95% CI [−1.60 to −0.28]; $I^2$ = 0%; $p$ = 0.009) with low-quality evidence compared to ST (Figs. 8–10, Table S3).

*Cardiorespiratory fitness*
CART was included in 16 trials, involving 420 participants, showing very low-quality evidence, and inducing a significant improvement in CRF (SMD 0.40, 95% CI [−0.02 to 0.77]; $I^2$ = 67%; $p$ = 0.008) compared to ST (Fig. 11, Table S3).

## Secondary outcomes
*Anthropometry*
BMI was reported in 16 trials ($n$ = 915) demonstrating low-quality evidence and exerting a meaningful reduction in BMI (SMD −0.32, 95% CI [−0.57 to −0.08]; $I^2$ = 65%; $p$ = 0.001) compared to ST (Fig. 12, Table S3).

*Quality of life*
QoL was investigated in three trials involving 358 participants and showing low-quality evidence. CART improved QoL (SMD 0.29, 95% CI [0.03–0.56]; $I^2$ = 37%; $p$ = 0.03) (Fig. 13, Table S3) compared to ST.

## DISCUSSION
In the present study we provide, for the first time to our knowledge, evidence about the effectiveness of CART on several cardiometabolic health-related indices. The main findings suggest that CART induces positive changes in glycemic control, blood pressure, inflammation, cardiorespiratory fitness, as well as quality of life in patients with T2DM and concurrent overweight/obesity. Considering that aerobic and resistance training alone have been reported as effective exercise solutions for inducing favorable results in cardiometabolic health among people with T2DM (*Al-Mhanna et al., 2023*; *Grace et al., 2017*; *Kelley & Kelley, 2007*; *Nery et al., 2017*; *Yang et al., 2014*), the present findings

**Figure 3 Risk of bias assessment results.** Note: *Annibalini et al. (2017)*, *Church et al. (2010)*, *Cuff et al. (2003)*, *Dunstan et al. (1998)*, *Earnest et al. (2014)*, *Ferrer-García et al. (2011)*, *Gibbs et al. (2012)*, *Hale et al. (2022)*, *Jennings et al. (2009)*, *Jorge et al. (2011)*, *Lambers et al. (2008)*, *Loimaala et al. (2003)*, *Magalhaes et al. (2020, 2019)*, *Maiorana et al. (2001, 2002)*, *Oliveira et al. (2012)*, *Reid et al. (2010)*, *Sabouri et al. (2021)*, *Scheer et al. (2020)*, *Sigal et al. (2007)*, *Swift et al. (2012)*, *Tan, Li & Wang (2012)*, *Tessier et al. (2000)*, *Yavari et al. (2012)*, *Zarei et al. (2021)*.

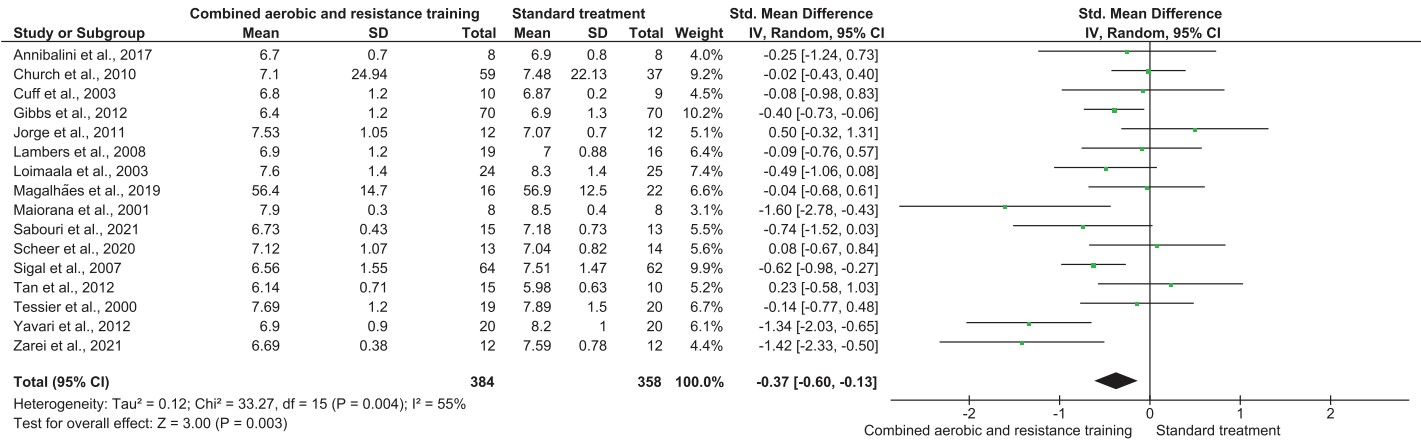

| Study or Subgroup | Combined aerobic and resistance training Mean | SD | Total | Standard treatment Mean | SD | Total | Weight | Std. Mean Difference IV, Random, 95% CI |
|---|---|---|---|---|---|---|---|---|
| Annibalini et al., 2017 | 6.7 | 0.7 | 8 | 6.9 | 0.8 | 8 | 4.0% | -0.25 [-1.24, 0.73] |
| Church et al., 2010 | 7.1 | 24.94 | 59 | 7.48 | 22.13 | 37 | 9.2% | -0.02 [-0.43, 0.40] |
| Cuff et al., 2003 | 6.8 | 1.2 | 10 | 6.87 | 0.2 | 9 | 4.5% | -0.08 [-0.98, 0.83] |
| Gibbs et al., 2012 | 6.4 | 1.2 | 70 | 6.9 | 1.3 | 70 | 10.2% | -0.40 [-0.73, -0.06] |
| Jorge et al., 2011 | 7.53 | 1.05 | 12 | 7.07 | 0.7 | 12 | 5.1% | 0.50 [-0.32, 1.31] |
| Lambers et al., 2008 | 6.9 | 1.2 | 19 | 7 | 0.88 | 16 | 6.4% | -0.09 [-0.76, 0.57] |
| Loimaala et al., 2003 | 7.6 | 1.4 | 24 | 8.3 | 1.4 | 25 | 7.4% | -0.49 [-1.06, 0.08] |
| Magalhães et al., 2019 | 56.4 | 14.7 | 16 | 56.9 | 12.5 | 22 | 6.6% | -0.04 [-0.68, 0.61] |
| Maiorana et al., 2001 | 7.9 | 0.3 | 8 | 8.5 | 0.4 | 8 | 3.1% | -1.60 [-2.78, -0.43] |
| Sabouri et al., 2021 | 6.73 | 0.43 | 15 | 7.18 | 0.73 | 13 | 5.5% | -0.74 [-1.52, 0.03] |
| Scheer et al., 2020 | 7.12 | 1.07 | 13 | 7.04 | 0.82 | 14 | 5.6% | 0.08 [-0.67, 0.84] |
| Sigal et al., 2007 | 6.56 | 1.55 | 64 | 7.51 | 1.47 | 62 | 9.9% | -0.62 [-0.98, -0.27] |
| Tan et al., 2012 | 6.14 | 0.71 | 15 | 5.98 | 0.63 | 10 | 5.2% | 0.23 [-0.58, 1.03] |
| Tessier et al., 2000 | 7.69 | 1.2 | 19 | 7.89 | 1.5 | 20 | 6.7% | -0.14 [-0.77, 0.48] |
| Yavari et al., 2012 | 6.9 | 0.9 | 20 | 8.2 | 1 | 20 | 6.1% | -1.34 [-2.03, -0.65] |
| Zarei et al., 2021 | 6.69 | 0.38 | 12 | 7.59 | 0.78 | 12 | 4.4% | -1.42 [-2.33, -0.50] |
| **Total (95% CI)** | | | **384** | | | **358** | **100.0%** | **-0.37 [-0.60, -0.13]** |

Heterogeneity: Tau² = 0.12; Chi² = 33.27, df = 15 (P = 0.004); I² = 55%
Test for overall effect: Z = 3.00 (P = 0.003)

**Figure 4  The effect of CART on HbA1c (primary outcome).** Note: *Annibalini et al. (2017)*, *Church et al. (2010)*, *Cuff et al. (2003)*, *Gibbs et al. (2012)*, *Jorge et al. (2011)*, *Lambers et al. (2008)*, *Loimaala et al. (2003)*, *Magalhães et al. (2019)*, *Sabouri et al. (2021)*, *Scheer et al. (2020)*, *Sigal et al. (2007)*, *Tan, Li & Wang (2012)*, *Tessier et al. (2000)*, *Yavari et al. (2012)*, *Zarei et al. (2021)*.     

| Study or Subgroup | Combined aerobic and resistance training Mean | SD | Total | Standard treatment Mean | SD | Total | Weight | Std. Mean Difference IV, Random, 95% CI |
|---|---|---|---|---|---|---|---|---|
| Loimaala et al., 2003 | 65.7 | 8 | 24 | 68.1 | 9 | 25 | 62.1% | -0.28 [-0.84, 0.29] |
| Maiorana et al., 2001 | 66 | 3 | 8 | 70 | 3 | 8 | 37.9% | -1.26 [-2.36, -0.16] |
| **Total (95% CI)** | | | **32** | | | **33** | **100.0%** | **-0.65 [-1.59, 0.29]** |

Heterogeneity: Tau² = 0.28; Chi² = 2.43, df = 1 (P = 0.12); I² = 59%
Test for overall effect: Z = 1.36 (P = 0.17)

**Figure 5  The effect of CART on RHR (primary outcome).** Note: *Loimaala et al. (2003)*, *Maiorana et al. (2001)*.

| Study or Subgroup | Combined aerobic and resistance training Mean | SD | Total | Standard treatment Mean | SD | Total | Weight | Std. Mean Difference IV, Random, 95% CI |
|---|---|---|---|---|---|---|---|---|
| Annibalini et al., 2017 | 116.6 | 12.9 | 8 | 128.1 | 10.7 | 8 | 6.5% | -0.92 [-1.96, 0.13] |
| García et al., 2011 | 141.43 | 15.45 | 44 | 143.47 | 18.47 | 40 | 11.5% | -0.12 [-0.55, 0.31] |
| Gibbs et al., 2012 | 125 | 14 | 70 | 125 | 13 | 70 | 12.3% | 0.00 [-0.33, 0.33] |
| Hale et al., 2022 | 136.1 | 1.12 | 83 | 134.9 | 1.13 | 82 | 12.4% | 1.06 [0.74, 1.39] |
| Jorge et al., 2011 | 125 | 13 | 10 | 124.2 | 17.3 | 12 | 8.0% | 0.05 [-0.79, 0.89] |
| Loimaala et al., 2003 | 138 | 16 | 24 | 144 | 14 | 25 | 10.3% | -0.39 [-0.96, 0.17] |
| Sabouri et al., 2021 | 125.2 | 6.13 | 15 | 128.46 | 5.31 | 13 | 8.7% | -0.55 [-1.31, 0.21] |
| Sigal et al., 2007 | 129 | 23 | 64 | 129 | 21 | 62 | 12.2% | 0.00 [-0.35, 0.35] |
| Tan et al., 2012 | 130 | 19 | 15 | 137 | 18 | 10 | 8.3% | -0.36 [-1.17, 0.44] |
| Yavari et al., 2012 | 123 | 12.5 | 20 | 121.3 | 14.4 | 20 | 9.9% | 0.12 [-0.50, 0.74] |
| **Total (95% CI)** | | | **353** | | | **342** | **100.0%** | **-0.04 [-0.41, 0.32]** |

Heterogeneity: Tau² = 0.26; Chi² = 45.38, df = 9 (P < 0.00001); I² = 80%
Test for overall effect: Z = 0.24 (P = 0.81)

**Figure 6  The effect of CART on SBP (primary outcome).** Note: *Annibalini et al. (2017)*, *Ferrer-García et al. (2011)*, *Gibbs et al. (2012)*, *Hale et al. (2022)*, *Jorge et al. (2011)*, *Loimaala et al. (2003)*, *Sabouri et al. (2021)*, *Sigal et al. (2007)*, *Tan, Li & Wang (2012)*, *Yavari et al. (2012)*.

indicate that CART may be considered as the optimal exercise approach for populations with impaired metabolic health due to the concurrent presence of T2DM and overweight/obesity.

Resistance training has been shown to increase muscle mass and glycogen storage (*Knuiman, Hopman & Mensink, 2015*), whereas aerobic exercise improves mitochondrial function and glucose utilization (*Wang et al., 2020*). Together, they contribute to improved glycemic control by reducing insulin resistance and enhancing glucose metabolism (*Yavari et al., 2012*). Additionally, aerobic exercise enhances cardiac output, vascular function, and

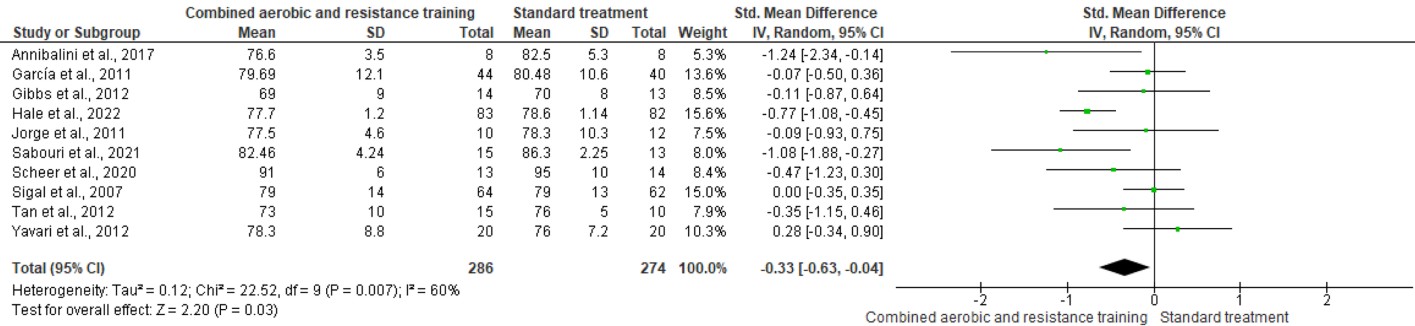

**Figure 7** **The effect of CART on DBP (primary outcome).** Note: *Annibalini et al. (2017)*, *Ferrer-García et al. (2011)*, *Gibbs et al. (2012)*, *Hale et al. (2022)*, *Jorge et al. (2011)*, *Sabouri et al. (2021)*, *Scheer et al. (2020)*, *Sigal et al. (2007)*, *Tan, Li & Wang (2012)*, *Yavari et al. (2012)*.

**Figure 8** **The effect of CART on CRP (primary outcome).** Note: *Annibalini et al. (2017)*, *Church et al. (2010)*, *Magalhães et al. (2019)*.

**Figure 9** **The effect of CART on TNF-α (primary outcome).** Note: *Annibalini et al. (2017)*, *Magalhães et al. (2019)*.

**Figure 10** **The effect of CART on IL-6 (primary outcome).** Note: *Annibalini et al. (2017)*, *Jorge et al. (2011)*.

endothelial health, while resistance training improves arterial stiffness and regulation of blood pressure (*Al-Mhanna et al., 2022b*; *Okamoto, Masuhara & Ikuta, 2013*). CART leverages these benefits, resulting in enhanced cardiovascular fitness and heart health. CART likely surpasses single-mode exercises due to its ability to capitalize on the distinct effects of both aerobic and resistance training, leading to more comprehensive improvements in cardiometabolic health indicators (*Jorge et al., 2011*). Therefore, the integration of CART represents a promising strategy for optimising cardiometabolic health in patients with T2DM and overweight/obesity. Its multifaceted benefits, including

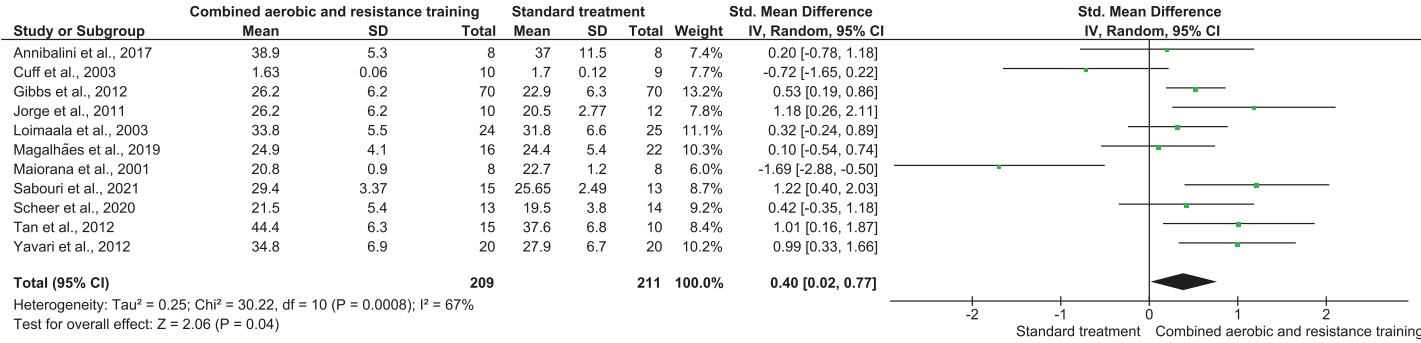

| Study or Subgroup | Combined aerobic and resistance training Mean | SD | Total | Standard treatment Mean | SD | Total | Weight | Std. Mean Difference IV, Random, 95% CI |
|---|---|---|---|---|---|---|---|---|
| Annibalini et al., 2017 | 38.9 | 5.3 | 8 | 37 | 11.5 | 8 | 7.4% | 0.20 [-0.78, 1.18] |
| Cuff et al., 2003 | 1.63 | 0.06 | 10 | 1.7 | 0.12 | 9 | 7.7% | -0.72 [-1.65, 0.22] |
| Gibbs et al., 2012 | 26.2 | 6.2 | 70 | 22.9 | 6.3 | 70 | 13.2% | 0.53 [0.19, 0.86] |
| Jorge et al., 2011 | 26.2 | 6.2 | 10 | 20.5 | 2.77 | 12 | 7.8% | 1.18 [0.26, 2.11] |
| Loimaala et al., 2003 | 33.8 | 5.5 | 24 | 31.8 | 6.6 | 25 | 11.1% | 0.32 [-0.24, 0.89] |
| Magalhães et al., 2019 | 24.9 | 4.1 | 16 | 24.4 | 5.4 | 22 | 10.3% | 0.10 [-0.54, 0.74] |
| Maiorana et al., 2001 | 20.8 | 0.9 | 8 | 22.7 | 1.2 | 8 | 6.0% | -1.69 [-2.88, -0.50] |
| Sabouri et al., 2021 | 29.4 | 3.37 | 15 | 25.65 | 2.49 | 13 | 8.7% | 1.22 [0.40, 2.03] |
| Scheer et al., 2020 | 21.5 | 5.4 | 13 | 19.5 | 3.8 | 14 | 9.2% | 0.42 [-0.35, 1.18] |
| Tan et al., 2012 | 44.4 | 6.3 | 15 | 37.6 | 6.8 | 10 | 8.4% | 1.01 [0.16, 1.87] |
| Yavari et al., 2012 | 34.8 | 6.9 | 20 | 27.9 | 6.7 | 20 | 10.2% | 0.99 [0.33, 1.66] |
| **Total (95% CI)** | | | 209 | | | 211 | 100.0% | 0.40 [0.02, 0.77] |

Heterogeneity: Tau² = 0.25; Chi² = 30.22, df = 10 (P = 0.0008); I² = 67%
Test for overall effect: Z = 2.06 (P = 0.04)

**Figure 11** **The effect of CART on CRF (primary outcome).** Note: *Annibalini et al. (2017)*, *Cuff et al. (2003)*, *Gibbs et al. (2012)*, *Jorge et al. (2011)*, *Loimaala et al. (2003)*, *Magalhães et al. (2019)*, *Sabouri et al. (2021)*, *Scheer et al. (2020)*, *Tan, Li & Wang (2012)*, *Yavari et al. (2012)*.

| Study or Subgroup | Combined aerobic and resistance training Mean | SD | Total | Standard treatment Mean | SD | Total | Weight | Std. Mean Difference IV, Random, 95% CI |
|---|---|---|---|---|---|---|---|---|
| **1.14.1 16 weeks or less** | | | | | | | | |
| Annibalini et al., 2017 | 27.7 | 2.1 | 8 | 29.1 | 3.6 | 8 | 4.0% | -0.45 [-1.45, 0.55] |
| Cuff et al., 2003 | 30.4 | 1.3 | 10 | 38.7 | 1.2 | 9 | 1.0% | -6.32 [-8.75, -3.89] |
| Dunstan et al., 1998 | 28.1 | 0.8 | 11 | 30.4 | 1.1 | 10 | 3.3% | -2.31 [-3.47, -1.16] |
| Lambers et al., 2008 | 28.5 | 2.85 | 19 | 30.3 | 4.34 | 16 | 6.1% | -0.49 [-1.16, 0.19] |
| Magalhães et al., 2019 | 30.8 | 5.4 | 16 | 31.1 | 5 | 22 | 6.3% | -0.06 [-0.70, 0.59] |
| Sabouri et al., 2021 | 25.89 | 2.15 | 15 | 26.4 | 2.6 | 13 | 5.5% | -0.21 [-0.95, 0.54] |
| Tessier et al., 2000 | 30.6 | 5.4 | 19 | 29.4 | 3.8 | 20 | 6.5% | 0.25 [-0.38, 0.88] |
| Zarei et al., 2021 | 26.17 | 3.94 | 13 | 27.3 | 2.36 | 13 | 5.3% | -0.34 [-1.11, 0.44] |
| Subtotal (95% CI) | | | 111 | | | 111 | 38.0% | -0.79 [-1.49, -0.08] |

Heterogeneity: Tau² = 0.79; Chi² = 38.88, df = 7 (P < 0.00001); I² = 82%
Test for overall effect: Z = 2.19 (P = 0.03)

| | | | | | | | | |
|---|---|---|---|---|---|---|---|---|
| **1.14.2 More than 16 weeks** | | | | | | | | |
| Church et al., 2010 | 34.3 | 16.16 | 76 | 35.2 | 10.3 | 41 | 8.8% | -0.06 [-0.44, 0.32] |
| García et al., 2011 | 30.37 | 5.86 | 44 | 32.35 | 7.22 | 40 | 8.3% | -0.30 [-0.73, 0.13] |
| Gibbs et al., 2012 | 31.6 | 4.9 | 70 | 33.4 | 4.4 | 70 | 9.2% | -0.38 [-0.72, -0.05] |
| Loimaala et al., 2003 | 28.9 | 3.8 | 24 | 29.8 | 3.7 | 25 | 7.1% | -0.24 [-0.80, 0.33] |
| Maiorana et al., 2001 | 34.2 | 9.6 | 64 | 34.9 | 8.7 | 62 | 9.1% | -0.08 [-0.43, 0.27] |
| Sigal et al., 2007 | 34.2 | 9.6 | 64 | 34.9 | 8.7 | 62 | 9.1% | -0.08 [-0.43, 0.27] |
| Tan et al., 2012 | 25.3 | 2.7 | 15 | 25.8 | 2 | 10 | 5.1% | -0.20 [-1.00, 0.61] |
| Yavari et al., 2012 | 26.17 | 3.94 | 13 | 27.3 | 2.36 | 13 | 5.3% | -0.34 [-1.11, 0.44] |
| Subtotal (95% CI) | | | 370 | | | 323 | 62.0% | -0.19 [-0.34, -0.04] |

Heterogeneity: Tau² = 0.00; Chi² = 2.97, df = 7 (P = 0.89); I² = 0%
Test for overall effect: Z = 2.46 (P = 0.01)

| | | | | | | | | |
|---|---|---|---|---|---|---|---|---|
| **Total (95% CI)** | | | 481 | | | 434 | 100.0% | -0.32 [-0.57, -0.08] |

Heterogeneity: Tau² = 0.14; Chi² = 43.22, df = 15 (P = 0.0001); I² = 65%
Test for overall effect: Z = 2.56 (P = 0.01)
Test for subgroup differences: Chi² = 2.64, df = 1 (P = 0.10), I² = 62.1%

**Figure 12** **The effect of CART on BMI (secondary outcome).** Note: *Annibalini et al. (2017)*, *Church et al. (2010)*, *Cuff et al. (2003)*, *Dunstan et al. (1998)*, *Ferrer-García et al. (2011)*, *Gibbs et al. (2012)*, *Lambers et al. (2008)*, *Loimaala et al. (2003)*, *Magalhães et al. (2019)*, *Maiorana et al. (2001)*, *Sabouri et al. (2021)*, *Sigal et al. (2007)*, *Tan, Li & Wang (2012)*, *Tessier et al. (2000)*, *Yavari et al. (2012)*, *Zarei et al. (2021)*.

| Study or Subgroup | Combined aerobic and resistance training Mean | SD | Total | Standard treatment Mean | SD | Total | Weight | Std. Mean Difference IV, Random, 95% CI |
|---|---|---|---|---|---|---|---|---|
| García et al., 2011 | 0.76 | 0.35 | 44 | 0.62 | 0.37 | 40 | 26.8% | 0.39 [-0.05, 0.82] |
| Hale et al., 2022 | 0.839 | 0.248 | 83 | 0.82 | 0.243 | 82 | 41.4% | 0.08 [-0.23, 0.38] |
| Sigal et al., 2007 | 28.6 | 5.4 | 57 | 25.9 | 5.4 | 52 | 31.7% | 0.50 [0.11, 0.88] |
| **Total (95% CI)** | | | 184 | | | 174 | 100.0% | 0.29 [0.03, 0.56] |

Heterogeneity: Tau² = 0.02; Chi² = 3.16, df = 2 (P = 0.21); I² = 37%
Test for overall effect: Z = 2.15 (P = 0.03)

**Figure 13** **The effect of CART on QoL (secondary outcome).** Note: *Ferrer-García et al. (2011)*, *Hale et al. (2022)*, *Sigal et al. (2007)*.

enhanced glycemic control, body composition, and cardiovascular function, underscore its potential as a cornerstone of exercise-based interventions in clinical settings. However, certain studies included in our review exhibited varying levels of bias, ranging from unclear to high risk. The presence of bias in these studies could potentially influence the overall conclusions drawn from our analysis. For instance, studies with a high risk of bias might overestimate or underestimate certain effects, thus affecting the generalizability of our findings. Additionally, unclear bias in key studies can introduce uncertainty into our assessment of the true effect sizes and outcomes being studied.

### Glycemic control

According to our findings, CART showed a meaningful reduction in HbA1c among people with T2DM and concurrent overweight/obesity. These findings indicated that CART likely improves glycemic control through enhanced insulin sensitivity, increased muscle glucose uptake, and favourable changes in body composition, including reduced adiposity. These combined effects contribute to improved glucose metabolism and overall glycemic management in individuals with T2DM and overweight/obesity.

It is worth mentioning that significant reductions in HbA1c result in a decreased risk of developing diabetes-related mortality. This outcome corroborates results reported in previous meta-analyses investigating the impact of physical exercise on glucose metabolism among populations affected by T2DM with or without concurrent overweight/obesity (*Zhao et al., 2021*). Interestingly, long-term (>12 weeks) exercise interventions appear to be more effective than short-term (≤12 weeks), underlining the positive role of physical exercise in glycemic management among patients with T2DM and obesity in a prolonged way (*Church et al., 2010*; *Zou et al., 2016*). In addition, CART have been documented as the optimal exercise strategy for inducing beneficial alterations in glycemic control among people with metabolic health impairments compared to other training modalities, such as aerobic, resistance, or interval training (*Batrakoulis, Jamurtas & Fatouros, 2021*; *Batrakoulis et al., 2022a*; *Zhao et al., 2021*). In summary, the beneficial role of CART in glycemic management may be partly explained by the fact that abdominal obesity is common among populations with T2DM and excessive weight (*Bays et al., 2008*). Thus, such CART-induced adaptations may be linked to the potential activation of some key molecular mechanisms responsible for regulating whole-body glucose homeostasis associated with visceral adipose tissue (*Magalhães et al., 2020*; *Sabag et al., 2017*).

### Blood pressure

Populations with T2DM and concurrent overweight/obesity are at high risk of experiencing raised blood pressure, resulting in a high risk of cardiovascular disease morbidity and mortality (*Costanzo et al., 2015*). According to the American Diabetic Association's guidelines, it is important for these populations to maintain normal blood pressure levels in order to lower the risk of developing metabolic syndrome (*Colberg et al., 2016*). In the present study, CART exerted a substantial reduction in DBP; however, no beneficial changes were observed in SBP and RHR. In general, resting cardiovascular

function improvements in individuals with T2DM are not well defined. More specifically, conflicting results are present when examining the role of CART in blood pressure among people with impaired glycemic control and with or without overweight/obesity (*Albalawi et al., 2017*; *Bersaoui et al., 2020*; *Zhao et al., 2021*). This remark cannot be explained here; however, the presence of overweight/obesity along with T2DM may play some role in the simultaneous management of HbA1c and blood pressure due to a complex, low-grade chronic inflammation. Moreover, CART may not be the optimal exercise solution for lowering resting cardiovascular function parameters, since it is characterized by high weekly time commitment (210–270 min) (*Larose et al., 2011*). This is important, considering how stressful CART may be compared to other less intensive and more time-efficient exercise types (*Batrakoulis, 2022a*, *2022b*, *2022c*; *Batrakoulis & Fatouros, 2022*). However, an Alternative exercise modality that could be considered other exercise types beyond CART is High-Intensity Interval Training (HIIT) which may offer distinct advantages in this context. HIIT involves alternating short bursts of intense exercise with recovery periods. Studies have indicated that HIIT can effectively reduce blood pressure in individuals with T2DM and overweight/obesity (*Ahmad et al., 2023*; *de Oliveira Teles et al., 2022*). The combination of aerobic and anaerobic components in HIIT may confer unique cardiovascular benefits.

Future research should focus on directly comparing the efficacy of these alternative exercise modalities to inform evidence-based recommendations.

## Inflammation

Individuals with impaired metabolic health are likely to present with raised inflammatory markers since a progressive accumulation of triglycerides promotes the fat cells hyperplasia and hypertrophy, resulting in a pro-inflammatory state which stimulates the generation of reactive oxygen species (*Bays et al., 2008*). CART with or without caloric restriction has been widely documented as an effective exercise strategy for populations with obesity-related chronic systemic inflammation (*Brunelli et al., 2015*; *Ihalainen et al., 2018*; *Lopes et al., 2016*; *Magalhães et al., 2020*; *Mendes et al., 2016*). In the current study, we found substantial CART-induced improvements in TNF-α and IL-6, but not in CRP. Such significant reductions are critical, given that people with T2DM and obesity tend to have various cardiovascular complications associated with elevated oxidative stress, impaired antioxidant capacity, insulin resistance and declined CRF due to chronic inflammation of adipose tissue by stimulating the immune system (*Bays et al., 2008*). Considering that both aerobic and resistance training alone induce favorable alterations in inflammatory markers among populations with T2DM and/or obesity (*Batrakoulis et al., 2021a*; *Batrakoulis, Jamurtas & Fatouros, 2021*; *Colberg et al., 2010a*; *Kanaley et al., 2022*; *Libardi et al., 2012*; *Qadir et al., 2021*), CART appears to be the optimal exercise approach for those characterized by metabolic health impairments (*Batrakoulis et al., 2022a*). The integration of aerobic and resistance exercise elicits potent anti-inflammatory effects by modulating adipose tissue metabolism and reducing pro-inflammatory cytokines. This comprehensive reduction in systemic inflammation is critical for mitigating cardiovascular risk factors associated with obesity and T2DM (*Scheffer & Latini, 2020*).

Taking these remarks into account, our findings indicate strong evidence considering the CART-induced beneficial changes in obesity-related inflammation, playing a vital role in lowering various cardiovascular disease risk factors in patients with T2DM and concurrent overweight/obesity.

## Cardiorespiratory fitness

People with metabolic health complications tend to demonstrate poor CRF levels and low functionality, resulting in high risk for cardiovascular disease morbidity (*Colberg et al., 2010b*; *Jensen et al., 2014*). On the contrary, high CRF levels are linked to low risk of all-cause mortality and morbidity in populations with no health problems and this has been reported as a more influential factor for overall health compared to anthropometric and body composition measurements (*McAuley et al., 2016*). Additionally, CRF was inversely correlated with ectopic fat accumulation and glucose intolerance; however, no statistically significant differences were found between people with T2DM and those without T2DM in respect of CRF levels (*Sabag et al., 2021*). As for the positive role of exercise in CRF, CART is considered the most effective exercise type for increasing CRF levels in adults with excessive weight and no comorbidities (*Batrakoulis et al., 2022a*; *O'Donoghue et al., 2021*). Interestingly, training regimens incorporating aerobic and muscle-strengthening activities into a single session appear to be productive in terms of physical fitness improvements among previously inactive individuals with an unhealthy weight (*Batrakoulis et al., 2018*, *2022b*, *2021b*). Such an important elevation in CRF may be evidenced by the CART-induced beneficial mitochondrial adaptations as well as the rises in skeletal muscle capillarization and oxidative metabolism in response to CART (*Murlasits, Kneffel & Thalib, 2018*). CART is a strategic exercise approach that elicits profound physiological adaptations, particularly enhancing cardiorespiratory health through multiple pathways. Firstly, CART optimises mitochondrial function within skeletal muscle cells. Mitochondria are the cellular powerhouses responsible for producing energy, and CART training stimulates mitochondrial biogenesis and efficiency, leading to improved oxidative metabolism (*Tan et al., 2023*; *Tucker et al., 2022*).

Additionally, CART promotes muscle capillarization, which refers to the growth of tiny blood vessels (capillaries) around muscle fibres. This increased capillarization enhances oxygen delivery to the muscles during exercise, supporting aerobic metabolism and endurance capacity (*Moro et al., 2019*). CART also induces specific adaptations that synergistically enhance both strength and cardiovascular fitness.

## Anthropometry

Our study shows that CART reduces BMI in individuals with T2DM and excessive weight, indicating potential improvements not only in other anthropometric parameters but also in various body composition measurements that need to be investigated in-depth in the future. Considering that weight loss and weight loss maintenance are challenging goals for people with impaired metabolic health, the present outcome seems to be important for these populations characterized by visceral adiposity. Interestingly, CART has been reported as the number exercise option for improving body mass, body fat percentage, fat

and fat-free mass, waist circumference and waist-to-hip ratio in adults with overweight/ obesity (*Batrakoulis et al., 2018*, *2022a*). Previous research also demonstrated similar effects of CART on several anthropometric and body composition indicators among populations with T2DM and concurrent overweight/obesity (*Pan et al., 2018*; *Zhao et al., 2021*). However, further RCTs are needed to determine whether CART can induce favorable alterations in abdominal adiposity that is linked to lower morbidity and mortality risks (*Mulligan et al., 2019*).

## Quality of life

It has been well documented that persons with excessive weight are very likely to present with poor mental health due to body dissatisfaction (*Gilyana, Batrakoulis & Zisi, 2023*) associated with insufficient physical activity levels (*Chekroud et al., 2018*), resulting in high attrition rates when engaging in exercise interventions (*Burgess et al., 2017*). In general, regular exercise is considered an effective solution for elevating QoL and body satisfaction (*Campbell & Hausenblas, 2009*) and mitigating the association between obesity and psychiatry illness (*Taylor et al., 2013*). Importantly, exercise protocols integrating aerobic and resistance exercises into a multicomponent training program demonstrate substantially high adherence among individuals with overweight/obesity (*Batrakoulis et al., 2020*). The present study shows that CART increases QoL in people with T2DM and concurrent overweight/obesity, encouraging this population to maintain high energy and low stress levels while lowering potential depressive and anxiety symptoms that are responsible for sedentarism and exercise amotivation (*Posthouwer et al., 2005*).

## Implications for future research

Given than CART seems to be a beneficial training modality for people with T2DM and concurrent overweight/obesity with respect to improvements in several cardiometabolic health-related indices, there is lack of robust evidence on the implementation of CART in the real world. Despite the current exercise recommendations for individuals with T2DM (*Aschner, 2017*; *Colberg et al., 2010b*; *Mendes et al., 2016*), further investigation is needed to determine the ideal training parameters, such as frequency, intensity and time in order to help clinicians and practitioners prescribe evidence-based, CART-like protocols to persons with T2DM and an unhealthy weight (*Batrakoulis, 2022a*). Likewise, future research should focus on the examination of the dose-response relationship between CART and cardiometabolic health outcomes not only in supervised, lab-based trials but also in a free-living environment as previously articulated (*Batrakoulis et al., 2019*). Such a research approach would support the practicability of CART, indicating whether one of the most comprehensive and popular exercise approaches (*A'Naja et al., 2024*) can be implemented for individuals with the most prevalent metabolic health impairments under real-world conditions.

In general, our results corroborate the current exercise prescription guidelines for various general and clinical populations (*Burtscher, Millet & Burtscher, 2023*; *Donnelly et al., 2009*; *Marwick et al., 2009*). However, future research should systematically explore various frequencies, intensities, and durations of CART to determine the most effective

protocols for improving metabolic health. This evidence-based approach will enable clinicians and health experts to confidently design tailored CART-like exercise programs that optimize outcomes for patients with T2DM and concurrent overweight/obesity. Further investigation into optimal training parameters will bridge the gap between research findings and practical implementation in clinical settings, enhancing the precision and impact of exercise interventions for this population.

### Limitations

The present meta-analysis has few limitations and thus the outcomes should be considered with caution. In the current study, we applied the GRADE approach, which provides a structured method for assessing the quality of evidence, which is crucial for drawing reliable conclusions. However, given the low and very low certainty of several outcomes, caution is needed when interpreting the results. Eligible studies demonstrated inconsistency with respect to the training parameters applied during the interventions, resulting in significant heterogeneity among the included trials. Our study shows that favorable CART-induced adaptations are existent primarily among middle-aged and older adults (mean age: 57 ± 7 years). Thus, present findings cannot be generalized to other age groups, such as young adults with T2DB and an unhealthy weight. Given the included outcome measures, the role of CART in cardiometabolic health of this particular population still remains unclear due to the lack of data in terms of body composition, lipid homeostasis, oxidative stress and physical function.

## CONCLUSIONS

The present systematic review and meta-analysis provides important insights into the implementation of CART for patients with T2DM and concurrent overweight/obesity as a component of a comprehensive management and treatment plan in a clinical setting. The outcomes indicate clear evidence that CART has a positive role in improving key cardiometabolic and mental health-related indicators, such as glycemic control, blood pressure, chronic inflammation, cardiorespiratory fitness, and quality of life in patients with T2DM and concurrent overweight/obesity. More studies with robust methodological design are warranted to examine the dose-response relationship, training parameters configuration and mechanisms behind these positive adaptations. This review also underlines the need for further RCTs to investigate more comprehensive anthropometric and body composition outcome measures to intricate the CART-induced effects for individuals with T2DM and concurrent overweight/obesity.

### Funding

This work was supported by the Universiti Sains Malaysia (Grant No. 304.PPSP.6315639). The funders had no role in study design, data collection and analysis, decision to publish, or preparation of the manuscript.

## Grant Disclosures

The following grant information was disclosed by the authors:
Universiti Sains Malaysia: 304.PPSP.6315639.

## Competing Interests

The authors declare that they have no competing interests.

## Author Contributions

- Sameer Badri AL-Mhanna conceived and designed the experiments, performed the experiments, analyzed the data, prepared figures and/or tables, authored or reviewed drafts of the article, and approved the final draft.
- Alexios Batrakoulis conceived and designed the experiments, prepared figures and/or tables, authored or reviewed drafts of the article, and approved the final draft.
- Wan Syaheedah Wan Ghazali conceived and designed the experiments, performed the experiments, prepared figures and/or tables, authored or reviewed drafts of the article, and approved the final draft.
- Mahaneem Mohamed performed the experiments, analyzed the data, prepared figures and/or tables, authored or reviewed drafts of the article, and approved the final draft.
- Abdulaziz Aldayel conceived and designed the experiments, performed the experiments, prepared figures and/or tables, and approved the final draft.
- Maha H. Alhussain conceived and designed the experiments, performed the experiments, prepared figures and/or tables, and approved the final draft.
- Hafeez Abiola Afolabi performed the experiments, analyzed the data, prepared figures and/or tables, and approved the final draft.
- Yusuf Wada performed the experiments, analyzed the data, prepared figures and/or tables, and approved the final draft.
- Mehmet Gülü performed the experiments, analyzed the data, prepared figures and/or tables, authored or reviewed drafts of the article, and approved the final draft.
- Safaa Elkholi conceived and designed the experiments, prepared figures and/or tables, and approved the final draft.
- Bishir Daku Abubakar conceived and designed the experiments, prepared figures and/or tables, and approved the final draft.
- Daniel Rojas-Valverde conceived and designed the experiments, performed the experiments, analyzed the data, prepared figures and/or tables, authored or reviewed drafts of the article, and approved the final draft.

## Data Availability

    This is a systematic review/meta-analysis.

## Supplemental Information

Supplemental information for this article can be found online at http://dx.doi.org/10.7717/peerj.17525#supplemental-information.

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
