# Peer review of "Effects of combined aerobic and resistance training on glycemic control, blood pressure, inflammation, cardiorespiratory fitness and quality of life in patients with type 2 diabetes and overweight/obesity: a systematic review and meta-analysis"

_PeerJ, doi:10.7717/peerj.17525_

## Round 0.1 · original submission · Minor Revisions

Dear authors,

We kindly request that you carefully review the comments provided by the reviewers. Their valuable suggestions offer insights to enhance your manuscript. Incorporate their suggestions and carefully address all comments in your manuscript; it will significantly strengthen its content.

Thanks

Reviewer 1 ·

Basic reporting

1. Please increase the resolution of the figures (Figure 3-13). I cannot see the text clearly in the figure.
2. In line 204, the session number is missing.
3. I suggest the author provide the data as supplementary material for better reproducibility.

Experimental design

The author only analyzed several factor’s individual effects. I think there may be correlations across different factors. It will be more general to analyze the joint effect rather than the marginal effect only. I suggest the author do more analysis like principal component analysis to show which factors are more important.

Validity of the findings

The results are based on simply marginal analysis. More analysis across different factors is important for the readers to understand how these factors jointly affect the result.

Reviewer 3 ·

Basic reporting

Thank you to authors Sameer Badri AL-Mhanna et al. for investigating the effects of combined aerobic and resistance training (CART) on glycemic control, blood pressure, inflammation, cardiorespiratory fitness (CRF), and quality of life (QoL) in patients with type 2 diabetes (T2DM) and overweight/obesity. The aim of the study was to synthesize existing data to assess the effectiveness of CART in improving these health indicators. The study provides valuable insights. Here are some peer review comments on the study:

Abstract:
1.The Background and Methods sections are lengthy and could be condensed. In the Results section, it is recommended to provide more detailed explanations of the extent of improvement in each outcome measure due to CART, especially when significant differences are observed in the results.

Introduction:
1.While the authors mention the potential benefits of CART for patients with T2DM, a more detailed discussion on how CART affects the physiological mechanisms of cardiometabolic health-related indicators could be provided. For example, how does CART influence glycemic control, reduce inflammatory markers, or improve cardiovascular function?
2.Although the comparison between CART and single-form exercises is mentioned, further discussion on why CART may be superior to single aerobic or resistance training could be elaborated.
3.Ensure consistency in the use of professional terminology and definitions throughout the manuscript. For instance, "health disease" might be a typographical error; should it be corrected to "cardiovascular disease"?

Experimental design

Methods
1.The literature search strategy should detail the databases and search terms used. It is advisable to provide more details about the search strategy to ensure the robustness of the study. Specifically, Table S1 should list all the keywords used and their combinations to provide complete transparency of the search strategy.
2.The risk of bias assessment is thorough and follows established guidelines. However, some issues of unclear or high risk of bias in certain studies should be addressed in the discussion, emphasizing the potential impact on the study results.
3.Additionally, applying the GRADE approach provides a structured method for assessing the quality of evidence, which is crucial for drawing reliable conclusions. However, given the low and very low certainty in several outcomes, caution is needed when interpreting the results.

Validity of the findings

Results
1.“From the specified databases, PubMed, Web of Science, Scopus, Science Direct, Cochrane Library, and Google Scholar (Figure 1), a total of 21,612 studies were obtained. After removing duplicate articles, the number of studies eligible for further evaluation was reduced to 19,965. Through a review of the titles and abstracts based on predetermined inclusion and exclusion criteria, 19,926 studies were excluded. Subsequently, the full text of the remaining 39 articles was carefully examined, excluding 14 articles with reasons. Therefore, 25 records were included in this study. However, five records were subsequent studies of eligible trials included in this review. Hence, a total of 20 studies were finally included in this review, and data were extracted from 1,192 patients that met the eligibility criteria (Figure 1 and Tables S4 and S5).” The authors described the process of screening from 21,612 search results to 20 or 25 studies, but why are there 26 studies listed in Table S4? Additionally, there are many blank entries in the table. Is this because there were no relevant data included in the studies, or for another reason?
2. The geographical distribution and study settings of the included studies, such as high-income countries, upper-middle-income countries, etc., are described. How were these categories defined? Was it based on GDP per capita? Furthermore, providing more detailed descriptions of studies conducted in non-clinical settings would be beneficial for a more comprehensive understanding of the research context.
3. The authors described the main and secondary outcome measures separately, but it is advisable to label the titles beneath the figures. Additionally, improving the clarity of the figures is recommended.
Discussion
1.The authors provide compelling evidence regarding the impact of CART on glycemic control in patients with T2DM and overweight/obesity. However, it is suggested that the authors further elucidate the specific advantages of CART compared to single aerobic or resistance training in the discussion, as well as the specific guidance implications of these findings for clinical practice.
2.When explaining the impact of CART on HbA1c, it is recommended to provide more details about the physiological or molecular mechanisms by which CART may lead to these improvements. This would help deepen the readers' understanding of the study results.
3.It is suggested to discuss the potential effects of CART's high weekly time requirement on participants' engagement and sustainability, and how CART protocols could be optimized for practicality and effectiveness by adjusting exercise frequency, intensity, or duration.
4.The article mentions that CART may not be the optimal exercise solution for lowering blood pressure. It is recommended that the authors discuss other types of exercise that may be more suitable for patients with T2DM and overweight/obesity, and provide corresponding research evidence.
5.The article mentions that CART is an effective exercise strategy for improving obesity-related chronic systemic inflammation. It is suggested that the authors discuss the specific advantages of CART compared to other forms of exercise (such as pure aerobic exercise or resistance training), and the significance of these findings for formulating exercise interventions.
6.Provide more details and explanations on how CART enhances cardiorespiratory health by improving mitochondrial function, muscle capillarization, and oxidative metabolism.
7.Emphasize the need for future research to clarify the optimal training parameters of CART (such as frequency, intensity, and duration), so that clinicians and health experts can provide evidence-based CART-like exercise programs for patients with T2DM and concurrent overweight/obesity.

Annotated reviews are not available for download in order to protect the identity of reviewers who chose to remain anonymous.

Reviewer 4 ·

Basic reporting

The manuscript is well-structured, following standard reporting guidelines with clear, professional language. It adequately contextualizes the study with an extensive literature review, providing relevant citations.

Please ensure that each figure and table is clearly described within the text. Some figures and tables are referenced minimally, which might confuse readers about their relevance or the specifics of the data presented.

Experimental design

Data Extraction Process (Lines 153-158): It would be beneficial to elaborate on how conflicts were resolved during data extraction and to detail the double-checking process to ensure accuracy in the reported data.

Validity of the findings

Risk of Bias (Lines 160-167): The manuscript mentions assessing the risk of bias using the Cochrane Handbook but does not detail how these assessments influenced the analysis. Expanding on this point, possibly by explaining how high-risk studies were treated during meta-analysis, would add depth to the reliability of the conclusions.

Generalizability (Lines 191-196): The discussion could be enhanced by addressing the generalizability of the findings to populations beyond the included studies. A statement about how these results might differ in various demographic or geographical contexts would be useful.

Reviewer 5 ·

Basic reporting

This is an important and interesting study and nicely done where the researchers studied the effectiveness of combined aerobic and resistance training (CART) on glycemic control, blood pressure, inûammation, cardiorespiratory ûtness (CRF), and quality of life (QoL) in overweight and obese individuals with T2DM.

The authors have made some corrections based on the comments from the previous reviewers. Some more comments are as follows.

Abstract
Add the full form of T2DM when using it for the first time.

Introduction
The introduction is short and needs to be added more.
It is better to list important risk factors of DM type 2. And studies on various parameters such as glycemic control, blood pressure, inflammation, cardiorespiratory fitness (CRF), overweight, and obesity should be added. These will help in studying the role of aerobic and resistance training (CART) on these parameters. The authors can add a Table.

Materials and Methods
Line 204. Remove the typo.
Better to add from which year to which year the studies are included.
If possible, please add study selection criteria.

Discussion
Add future studies recommendations of the present study.

Results
The quality of the Figures is low. The figure’s quality needs to be improved for better visualization.

Experimental design

Better to add from which year to which year the studies are included.
If possible, please add study selection criteria.

Validity of the findings

-

Additional comments

-

---

## Round 0.2 · accepted · Accept

The authors have conducted excellent work on this manuscript, incorporating feedback from reviewers to enhance the clarity and depth of the content. They have successfully elucidated the combined effect of Combined Aerobic and Resistance Training on Glycemic Control, Blood Pressure, Inflammation, Cardiorespiratory Fitness in Patients with Type 2 Diabetes. Thanks to the thoughtful integration of the reviewers' suggestions.

Reviewer 1 ·

Basic reporting

The issues have been improved.

Experimental design

The issues have been improved.

Validity of the findings

The issues have been improved.

Reviewer 4 ·

Basic reporting

The authors made the corresponding in the revised version, the manuscript should be accepted.

Experimental design

no comment

Validity of the findings

no comment

Additional comments

no comment

Reviewer 5 ·

Basic reporting

The authors have improved the manuscript with the latest references and improved English throughout.

Experimental design

The experimental design is more clear.

Validity of the findings

The findings are presented in a better way.

Additional comments

It is better to add both full form and abbreviations in the figure legends. For example CART, DBP, TNF, etc.